

# IterCluster: a barcode clustering algorithm for long fragment read analysis

Jiancong Weng[1,2,*], Tian Chen[2,*], Yinlong Xie[2], Xun Xu[3], Gengyun Zhang[1], Brock A. Peters[3] and Radoje Drmanac[3]

[1] BGI Education Center, University of Chinese Academy of Sciences, Shenzhen, China
[2] MGI, BGI-Shenzhen, Shenzhen, China
[3] BGI-Shenzhen, Shenzhen, China
[*] These authors contributed equally to this work.

## ABSTRACT

Recent advances in long fragment read (LFR, also known as linked-read technologies or read-cloud) technologies, such as single tube long fragment reads (stLFR), 10X Genomics Chromium reads, and TruSeq synthetic long-reads, have enabled efficient haplotyping and genome assembly. However, in the case of stLFR and 10X Genomics Chromium reads, the long fragments of a genome are covered sparsely by reads in each barcode and most barcodes are contained in multiple long fragments from different regions, which results in inefficient assembly when using long-range information. Thus, methods to address these shortcomings are vital for capitalizing on the additional information obtained using these technologies. We therefore designed IterCluster, a novel, alignment-free clustering algorithm that can cluster barcodes from the same target region of a genome, using -mer frequency-based features and a Markov Cluster (MCL) approach to identify enough reads in a target region of a genome to ensure sufficient target genome sequence depth. The IterCluster method was validated using BGI stLFR and 10X Genomics chromium reads datasets. IterCluster had a higher precision and recall rate on BGI stLFR data compared to 10X Genomics Chromium read data. In addition, we demonstrated how IterCluster improves the de novo assembly results when using a divide-and-conquer strategy on a human genome data set (scaffold/contig N50 = 13.2 kbp/7.1 kbp vs. 17.1 kbp/11.9 kbp before and after IterCluster, respectively). IterCluster provides a new way for determining LFR barcode enrichment and a novel approach for de novo assembly using LFR data. IterCluster is OpenSource and available on https://github.com/JianCong-WENG/IterCluster.

# INTRODUCTION

The short read length of next-generation sequencing (NGS) technology presents a challenge for aligning programs in terms of how to handle the splicing of repeat sequences during the de novo assembly process. To address this problem, in addition to long-read sequencing, which provides reads that can often span across an entire repeat area, low-cost, low-input

Corresponding authors
Brock A. Peters,
bpeters@completegenomics.com
Radoje Drmanac,
rade@completegenomics.com

DNA library preparation techniques based on co-barcoding methods have been recently developed. These include technologies such as single tube long fragment reads (stLFR) from BGI, 10× Genomics, or TruSeq synthetic long reads (TSLR) from Illumina (*Wang et al., 2018*; *Zheng et al., 2016*; *Voskoboynik et al., 2013*). Using these techniques, a low-input DNA library is first trimmed into a series of long fragments of 10–100 kb in length and short reads from the same long fragment are created that all contain the same barcode. Since each fragment comes from a small fraction of the genome, the probability of a fragment containing identical repeat sequence is also low. Therefore, most of the repeat regions in the genome are covered by fragments with unique signatures. Once these are generated, the short reads are then sequenced using standard high-throughput sequencing technologies. The LFRs generated by these techniques are inseparable from the short read length features of NGS technology, but they provide more long-range information about the same DNA fragment compared to standard NGS short reads, thus providing more information for de novo genome assembly (*Weisenfeld et al., 2017*; *Coombe et al., 2018*; *Kuleshov, Snyder & Batzoglou, 2016*). In addition, LFR techniques have been used in haplotype phase determination and structural variation detection (*Wang et al., 2018*; *Zheng et al., 2016*).

To better explain the data characteristics of LFRs, three distinct coverage types for LFR data, similar to that proposed by Kuleshov, should be considered (*Kuleshov, Snyder & Batzoglou, 2016*). Specifically, these are local coverage, fragment coverage, and global coverage. Local coverage is the coverage of a long fragment with short reads in the same barcode and global coverage is the coverage of the whole genome with the sum of short reads from each fragment. The fragment coverage is the coverage of a whole genome with long fragments, which is directly related to the nature of the input DNA. In the case of BGI's stLFR technology, the number of cells used to make a 1 ng human DNA library equates to 300 haploid genomes, meaning the entire genome is covered at 300X depth in this library. Due to the limitation of sequencing cost, the local coverage of each fragment is very low (0.1–0.3) because it just needs to meet the experimental requirements of standard second-generation sequencing. The local coverage of each fragment is too low for each barcode's reads to completely contain fragment information, with the long-range information being sparse, and this low-sequencing depth cannot achieve the partial assembly of each fragment.

BGI's stLFR and 10× Genomics Chromium reads are similar technologies, but they have several distinct characteristics. They are both low-input (about 1 ng) DNA library preparation techniques, and each fragment from these two techniques generates the same number of reads. Compared with 10× Genomics Chromium reads, stLFRs rely more on a co-barcoding technique. stLFRs use the surface of microbeads to create millions of miniaturized compartments in a single tube and enable co-barcoding in reactions with 50 million barcodes. 10× Genomics Chromium reads can only utilize 1 million barcodes, so the number of fragments each barcode used with 10× Genomics Chromium reads are larger than the stLFR methodology at the same DNA-input level. For example, in a human genome library, the average fragments per barcodes is 1.18 using stLFR technology, but is

8.3 using 10× Genomics Chromium reads. The drawback, however, of stLFRs is their high duplicate rate at higher sequencing depths, which can impact downstream applications.

Unlike BGI's stLFR and 10× Genomics Chromium reads, TSLR technology guarantees the local coverage (10~30 X) of each fragment in a barcode by over-sequencing each barcode (*Voskoboynik et al., 2013*). Each barcode can be assembled separately using a short-read assembler and the assembly complexity is reduced because most of repetitive sequences become unique in a given long fragment. After this subassembly, the long sequences that have been assembled above can then be assembled to generate a whole genome using an overlap-layout-consensus (OLC) assembly strategy and this assembly strategy has been refined using several genomes (*Voskoboynik et al., 2013*; *McCoy et al., 2014*; *Li et al., 2015*). The low local coverage of BGI's stLFR data and 10× Genomics Chromium reads data make it impossible to achieve subassembly of each barcode and the de novo assembly algorithms designed specifically for BGI's stLFRs have not been made publically available.

Given a seed barcode representing a target genomic region, barcode clustering is an effective way to enhance the long-range information derived from LFR data and to enrich all of the reads and barcodes belonging to the same fragment region in the genome. The main purpose of barcode clustering is to capture all of the barcodes in a target genomic region and to filter out all of the reads from the region. In this way, the target region can be assembled independently. Barcode clustering separates the high complexity genome into relatively simple partitions, which reduces the de novo assembly complexity. However, the challenge for barcode clustering is that LFR technology groups reads from several fragment into the same barcode. For example, the data generated by BGI's stLFR technology contains roughly 1~3 long fragments per barcode, while data generated by 10× Genomics contains more fragments for each barcode, with an average of six barcodes per fragment. This complicates the problem of barcode clustering, as it causes clusters to generate a large number of false positive barcodes (Fig. 1C). In addition, if a seed barcode is contained in multiple fragments, the clustering eventually contains the barcodes from multiple fragment areas.

Recently, a novel algorithm, Minerva (*Danko et al., 2019*), provided a solution to these challenges. Minerva focuses on the barcode deconvolution problem utilizing a bipartite graph model. Minerva was designed to partition reads with a single barcode into clusters. Those clusters of reads are labeled with an enhanced barcode. After barcode deconvolution, the number of fragments containing an enhanced barcode can be reduced. Thus, the main purpose of barcode clustering and barcode deconvolution are quite different. Barcode deconvolution is aimed at polishing a single barcode, while barcode cluster focuses on capturing all the reads from a target genome region for helping de novo assembly. Unfortunately, Minerva is only available as a demo program and not very time-efficient. For example, it took more than 10 days to deconvolve a 50-fold human chromosome 19 dataset (https://figshare.com/articles/chr19_read1_fq_gz/7812038), and it only output 80,000 pairs of enhanced barcode reads using BGI's stLFR data. Thus, it is impossible for Minerva to finish clustering of a 50-fold whole genome sequencing dataset within an acceptable timeframe.

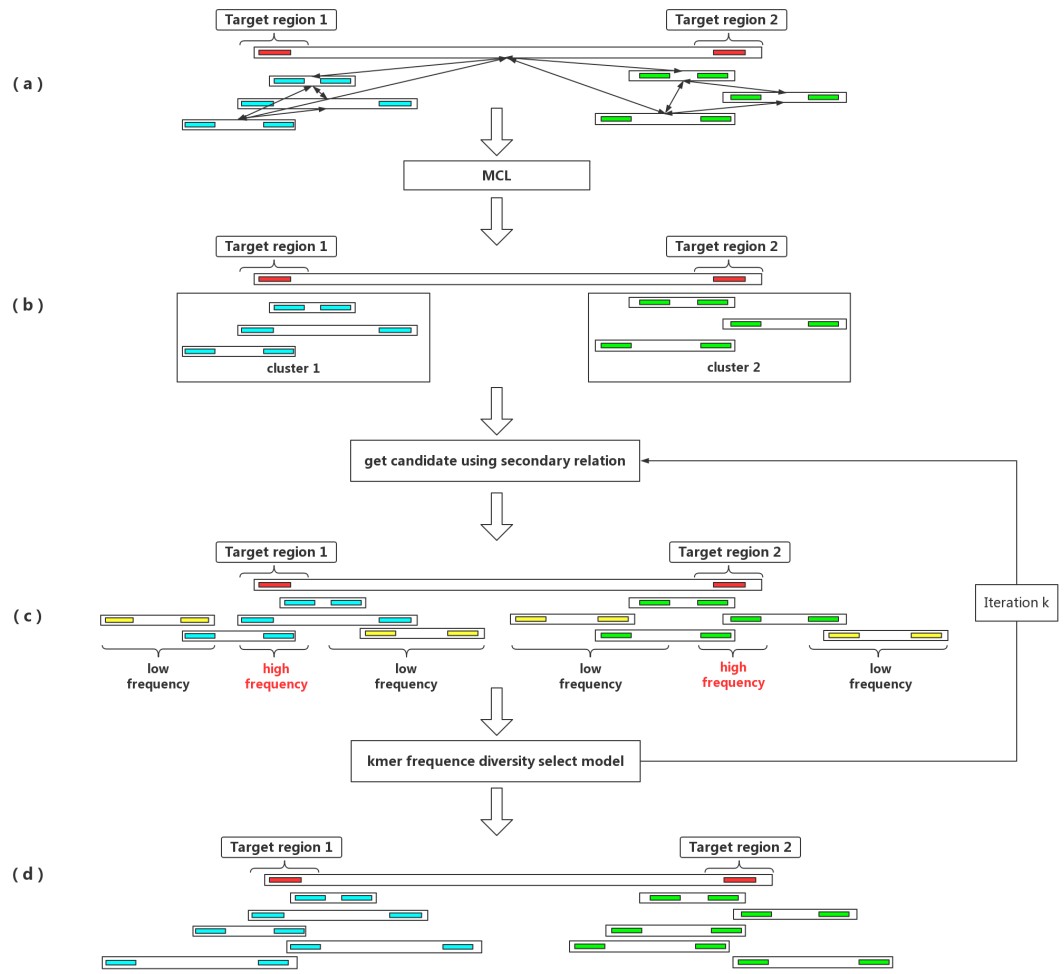

**Figure 1** **An outline of the IterCluster workflow.** (A) If the seed barcode contains two fragments (red bars), the cluster contains a sub-cluster from the two regions. The barcodes belonging to the same region are strongly connected, while barcodes from different regions are poorly connected. Markov clustering can achieve further division of the target barcode sub-cluster, so that each sub-cluster only contains barcodes coming from one target region. (B) Using a sub-cluster to get a candidate target barcode. (C) A target barcode directly captured by the seed barcode overlaps with the target region, but barcodes carrying multiple fragments lead to the introduction of a false positive fragment. If these sequences are used as seeds for the next capture, numerous barcodes (yellow bars) without overlap in the target region will be introduced. Using the difference in $k$-mer frequency between the target region and non-target regions, the high frequency unique $k$-mers is selected as the next captured feature value to control the false positive rate of barcodes. IterCluster uses an iterative capture, and each sub-cluster is independently enriched using the $k$-mer frequency diversity selection model to control the rate of false positive barcodes. High frequency unique $k$-mers are selected as seed features for the next round of enrichment, where the number of iterations $k$ is set by the user. (D) After $k$ iterations, each cluster can be obtained.

To address these limitations of using LFRs, we devised IterCluster, a novel reference- and alignment-free clustering algorithm that can be used to explore the potential relationships between barcodes and can cluster barcodes from the same target region of a genome. We developed a $k$-mer frequency diversity select model to control the false positives generated during barcode clustering, and have used a Markov clustering (MCL) model to ensure that

the clustering results only contain barcodes from one fragment area. IterCluster starts with a randomly selected seed barcode and is able to capture other barcodes with overlapping or nearby sequence identity to the seed barcode area. For each generated cluster we filter the non-target region's reads based on the reads depth difference, and we compared the performance of each cluster de novo assembled independently to all of the reads de novo assembled together in whole human genome data using stLFR technology. Also, we showed that IterCluster had a high recall rate and a suitable accuracy rate for human whole genome data. With IterCluster, we were able to achieve local assembly of a long fragment region, making the subassembly strategy possible in a reasonable timeframe.

## MATERIALS & METHODS

### IterCluster algorithm

Given a set of long-fragment reads, IterCluster selects appropriate barcodes as the seed and iteratively cluster target barcodes overlapping with the seed barcode. In order to minimize the overlap between clusters in the final clustering result, the target barcode that has become the seed barcode cannot be used as seed in future clusters. The size of each cluster depends on the number of iterations (parameter -k, default 3). The IterCluster algorithm proceeds in three steps: (1) constructing the adjacency matrix between barcodes; (2) generating barcode clusters; (3) extracting the reads of the target regions from each barcode cluster.

In the first step, we first extract the unique kmer in the genomic data set. The distance between the barcodes is measured by the number of common unique $k$-mers between the barcodes. In terms of time efficiency, IterCluster uses the sparsity of the overlap relationship between barcodes to generate the adjacency matrix.

In the second step, IterCluster evaluates the quality of the barcode by the unique kmer number and unique $k$-mers proportion, than adds the high quality barcode to the candidate seed list. IterCluster extracts the seed from the candidate seed list and clusters the seed barcode according to the adjacency matrix and the $k$-mers frequency diversity selection model to obtain the preliminary barcode cluster. Because most barcode carry fragments of multiple regions, the preliminary barcode cluster will also carry barcodes from multiple regions. IterCluster uses the Markov clustering model to disassemble each preliminary barcode cluster and obtain a single barcode cluster.

Finally, IterCluster will count the kmer coverage depth of the reads in each barcode cluster, identify the low coverage depth reads and filter out, and then get the reads in the barcode clustering area.

### Using unique $k$-mer as a similarity measurement

In the stLFR data, if there are overlaps between a long fragment contained in two barcodes, the read overlap in the two barcodes can be found, as these signify the similarity between barcodes. In measuring the similarity between barcodes, we did not use time-consuming methods such as read alignment between barcodes. Instead, we built a $k$-mer set for each barcode and measured the relation of barcodes based on the similarity of its $k$-mer set. Due to the fact that repeats are widely distributed in the genome, the $k$-mer set generated from a repeat would introduce a false-positive barcode. Therefore, we only select unique $k$-mers

as the feature of a barcode based on the $k$-mer frequency distribution. Let K (b) denote the unique $k$-mer set of barcode, the distance between barcodes can be described as:

Dist $(b_1, b_2) = K(b_1) \cap K(b_2)$.

## A time-efficient way to generate adjacency matrix

IterCluster implements barcode data clustering based on an adjacency matrix between barcodes. Many solutions to calculate the collective similarity exist, such as a local sensitive hash (*Charikar, 2002*), which has been applied to the de novo assembly of single-molecule sequencing data (*Berlin et al., 2015*) and the construction of phylogenetic trees (species trees) (*Ondov et al., 2016*). Although these techniques can efficiently detect the overlap between barcodes, a probabilistic method inevitably produces a certain error rate, and an absolute common unique $k$-mer number between barcodes cannot be calculated. To calculate the common unique $k$-mer number between barcodes, a simple, time-consuming method is to compute the intersection independently between every two barcodes and the computational complexity of this method is given as: $O(n^2)$.

IterCluster, however, utilizes an innovative algorithm to calculate the absolute common unique $k$-mer number between barcodes. The core logic behind this approach was that the adjacency matrix is sparse and each unique $k$-mer can determine which barcodes are related to each other. IterCluster extracts the unique $k$-mer of a barcode and builds a hash from the unique $k$-mer from each barcode. Barcodes that contain the same unique $k$-mer mean that the number of co-unique $k$-mers between them increases by one. Since there are few barcodes that simultaneously contain the same unique $k$-mer (theoretically less than or equal to the sequencing depth), it takes a constant time to accumulate the number of co-unique $k$-mers between the same unique $k$-mer numbers of supported barcodes. By traversing all of the unique $k$-mers, the absolute value of the co-unique $k$-mers between all of the barcodes can be calculated, and the time complexity of the program is given as: O $(n)$ (Fig. 2).

## Barcode enrichment based on a $k$-mer frequency diversity selection model

We developed an algorithm to iteratively cluster the barcodes from a target area in the genome. The reasoning behind this approach was that the barcodes covering a target area tend to contain unique $k$-mers derived from the target area while the content of unique $k$-mers derived from other areas should be random (Fig. 1C), since these barcode cover multiple long-fragment molecule. After each iteration of clustering, the unique $k$-mer frequency of target and non-target areas are significantly different. Using this difference to choose the unique $k$-mers belonging to a target area as a feature of the next round of clustering effectively controls the false-positive rate. IterCluster uses common unique $k$-mers to measure the similarity between barcodes. The number of common unique $k$-mers greater than the parameter **-g** between barcodes is considered to contain the overlapping long fragment molecule. In each iterative clustering process, IterCluster first captures the
| unique kmer | barcode ID | barcode ID | . . . . . | barcode ID |
|---|---|---|---|---|
| unique kmer k1 | barcode b1 | barcode b2 | . . . . . . | barcode b3 |
| unique kmer k2 | barcode b2 | barcode b4 | . . . . . . | barcode b1 |
| . . . . . . | . . . . . . | . . . . . . | . . . . . . | . . . . . . |
| unique kmer kn | barcode b100 | barcode 200 | . . . . . . | barcode 300 |

Co-unique kmer ( b1, b2 )  += 1
Co-unique kmer ( b2, b3 )  += 1
Co-unique kmer ( b1, b3 )  += 1

**Figure 2**  **Create a hash mapping from unique $k$-mer to barcode.**  Each row on the table above implies that there is relationship between those barcodes supported by a unique $k$-mer. Because barcodes have less probability of containing the same unique $k$-mer, the time complexity to calculate the relationship between each row of barcodes is O(1), and all row were traversed to calculate the relationship between all barcodes. The time Complexity is O(n).

**Table 1**  **Algorithm for barcode enrichment using a $k$-mer frequency selection model.**  $a$ is the seed barcode. $b$ represents all of the other barcodes. $T$ is the target barcode set. $F$ is a featured unique $k$-mer set with high frequency in each iteration. $Threshold$ is the relation threshold between barcodes, which is controlled by the $g$ parameter. $k_1$ is the number of iterations.

**Algorithm 1 IterCluster using $k$-mer frequency select model**

1. Generate seed barcode $a$
2. **For** each barcode($b$):
    3. if common unique $k$-mer ($a$, $b$) >$Threshold$:
        4. add $b$ to $T$
5. **End for**
6. Statistic unique $k$-mer frequency of reads in $T$, select the unique $k$-mer with high frequency to $F$
7. **For $k_1$** iteration:
    8. Get optional barcode set base on adjacency matrix
    9. **For** each barcode($b$) in optional barcode set
        10. if common unique $k$-mer ($b$, $F$) >$Threshold$:
        11. add $b$ to $T$
    12. Statistic unique $k$-mer frequency of reads in $T$
    13. Select unique $k$-mer with high frequency to $F$
        14. **End for**
15. **End for**

target barcodes directly overlapping the seed barcode and obtains the target barcode set $T$. Then it statistically determines the frequency of all of the unique $k$-mers of the barcode in $T$. The high-frequency unique $k$-mer set is then selected as the seed unique $k$-mer set for the next iteration and added to the feature $k$-mer set F (Table 1).

**Table 2  Algorithm for divide cluster by MCL.** $s$ is the matrix expansion coefficient (default is 2). $r$ is the matrix inflation coefficient (default is 2). $k$ is the column number of M. $p$ is each matrix row index. $q$ is each matrix column index. $k_2$ is the number of iterations.

**Algorithm 2 Divide cluster by MCL**

1. Given a target barcode set $T$, build adjacent matrix $M$
2. **For $k_2$ iteration:**
3. $M = M^s$
4. $M_{pq} = (\Gamma_r M)_{pq} = \dfrac{(M_{pq})^r}{\sum_{i=1}^{k}(M_{iq})^r}$

5. **End for**
6. extract sub cluster from $M$

## MCL model

Since each barcode covers multiple long fragment molecules (Fig. 1A), using a seed clustering strategy may result in a barcode set representing more than one target area in the genome. IterCluster uses a Markov clustering model (*Van Dongen, 2000*) to ensure that the result of seed clustering is a set of barcodes for individual areas (Fig. 1B). Specifically, assuming that a seed contains two long fragment molecules from different areas, the target barcode set will contain two clusters from different areas in the genome. However, the barcodes generated from the same area are closely related and barcodes generated from different area are distantly related. Therefore, IterCluster use a Markov clustering model to detect and classify closely-coupled barcode groups after the first clustering iteration (Table 2).

MCL algorithm has two main processes, expansion and inflation, which operate on the state transition matrix. When a state transition matrix is $M$, the dimension of $M$ is the number of points in the graph. Each list in $M$ represents the probability of starting from a certain point at a certain time and arriving at the remaining points at the next time.

The extended process is to simulate the random walk process. Taking positive integer $e$ and multiply the current state transition matrix by $e$ times to get a new state transition matrix, which is equivalent to a random walk on the original state transition matrix. This process can be described as:

$$M' = M^e.$$

The expansion process is a matrix regularization process, which regularizes the columns of the state transition matrix. The processing formula is shown as follow:

$$(M^*)_{pq} = \frac{(M_{pq})^T}{\sum_{i=1}^{K}(M_{iq})^T}.$$

Where $M$ is a state transition matrix; $M^\star$ is a regularized matrix; $T$ is a relaxation coefficient; $K$ is the number of rows of $M$; $P$ is a row subscript; $q$ is a column subscript. The function above is to regularize the columns of the transfer matrix to get the regularized matrix $M^\star$

**Table 3  Data set properties.**

|  | DataSet1 | DataSet2 |
|---|---|---|
| Technology | BGI's single tube LFR | 10× Genomics Chromium reads |
| Barcode number | $1.3 * 10^6$ | $3.6 * 10^7$ |
| Sequence depth | 56× | 50× |
| Fragment number of each | 1.18 | 8.32 |

For the $N$ target barcodes obtained after the seed barcode's first clustering iteration, the $N*N$ adjacency matrix $M$ was built and Markov clustering occurred. Finally, the sub-cluster can be found after MCL process.

## Data sources

We select 2 human genome data sets to evaluate IterCluster's performance: BGI stLFR data sets and 10× Genomics Chromium linked reads data sets (Table 3). BGI stLFR data were obtained for sample NA12878 from the CNGB Nucleotide Sequence Archive (CNSA) with accession ID: CNS0007594. The data were also available from the European Nucleotide Archive with accession ID: PRJEB27414.

Human Chromium linked reads data from 10× Genomics was obtained for a HGP sample, and were downloaded from the 10× Genomics company website (https://support.10Xgenomics.com/de-novo-assembly/datasets/2.1.0/hgp).

## Data analysis

### Barcode quality control

In the stLFR dataset, there has 52872872 barcode, but only 13630103 barcode have more than 10 pair reads, the small barcode can't find relationship with other barcode because of little unique $k$-mer. After filter small barcode which contain reads less than 10 pair, the barcodes' reads pair average count is 191. In the 10× Genomics dataset, there has 1394714 barcode, all of them are more than 10 reads pair; the average reads count is 490 for barcode. So it is not necessary to filter small barcode in 10× Genomics dataset.

### Split barcode and reads filter

We used our own pipeline to split the barcodes from reads and filtered the duplicate reads and adapters with SOAPfilter (v2.2), using the following settings: *-t* 30 *-q* 33 *-p* *-M* 2 *−y* *−F* adpter1 *−R* adapter2 *-f* *-1* *-Q* 10, obtaining a unique $k$-mer frequency range of 20∼80 by $k$-merfreq (in-house pipeline) with *−k* 17. The adapter sequence can be found in Supplemental Information. Then, we built the barcode's relation matrix, selecting an initial seed with a unique $k$-mer of (*−r* 0.3) and a $k$-mer count larger than 3000 (*-n* 3000), and then ran IterCluster with the following parameters: *-c* 10 *-k* 3 *-g* 200 *-f* 3 *-t* 80 *-a* 2 *-h* 12. Barcodes with read pair numbers less than 10 were deleted.

The barcodes from reads were extracted using Long Ranger (v2.1.2) and the duplicate reads were filtered using SOAPfilter (v2.2) with the settings *-t* 30 *-q* 33 *-p* *-M* 2 *-f* *-1* *-Q* 10. We obtained a unique $k$-mer frequency range 20∼60 by $k$-merfreq with *−k* 17. Then,

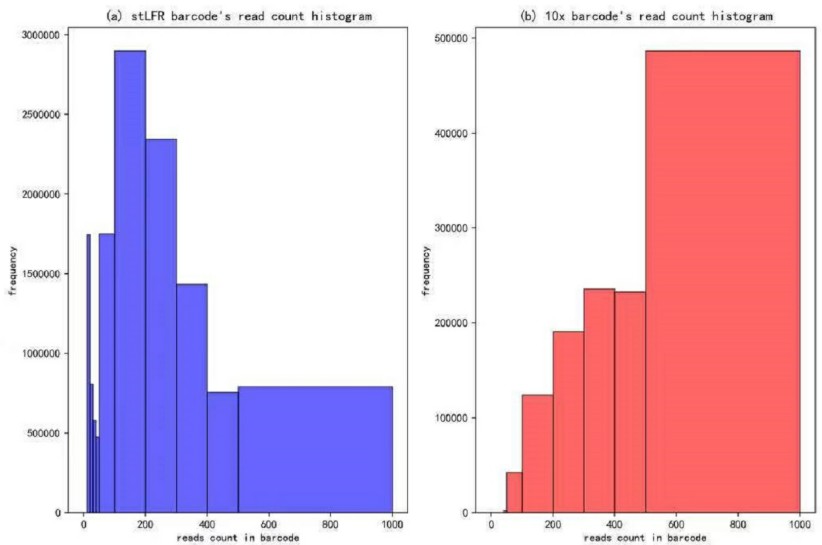

**Figure 3   Reads number distribution of each barcodes in BGI's stLFR and 10× Genomics dataset.** (A) stLRF barcode's read count histogram. (B) 10× barcode's read count histogram.

we built the barcode's relation matrix and ran IterCluster with the parameters *-c* 60 *-k* 2 *-n* 20000 *-r* 0.3 *-g* 130 *-f* 3 *-t* 45 *-a* 2 *-h* 12.

## Evaluation

In our IterCluster algorithm evaluation, we build actual positive dataset by aligning reads to genome reference with bwa (*Li & Durbin, 2009*) to know each barcode's position on the genomes. According to the range of LFR length and reads count in each barcode (Fig. 3), there are so many barcodes have little reads that haven't enough unique kmer to find other barcodes together. we defined initial seed barcodes that must cover areas where the reads spanned larger than 500 bp in length and had more than 30 paired-ends to represent an LFR fragment and the target barcodes that belong to an LFR fragment need to have more than 5 paired-ends, which can ensure the aggregation ability of the barcode.

After getting the cluster set, compare the IterCluster results with actual positive dataset, count the true positive barcode number that included by cluster set and actual target region barcode set, false positive barcode number that included by cluster set but actual target region barcode set in every random initial seeds' cluster set target region. And the span region for every initial seed may extend during IterCluster, we decided extend 80 kp∼400 kbp both left and right for the initial seed span region on reference as target region to evaluate. Using a P-R curve to evaluate, determining the Precision and Recall as follows:

$$Precision = \frac{True\ Positive}{True\ Positive + False\ Positive}.$$

$$Recall = \frac{True\ Positive}{True\ Positive + False\ Negative}.$$
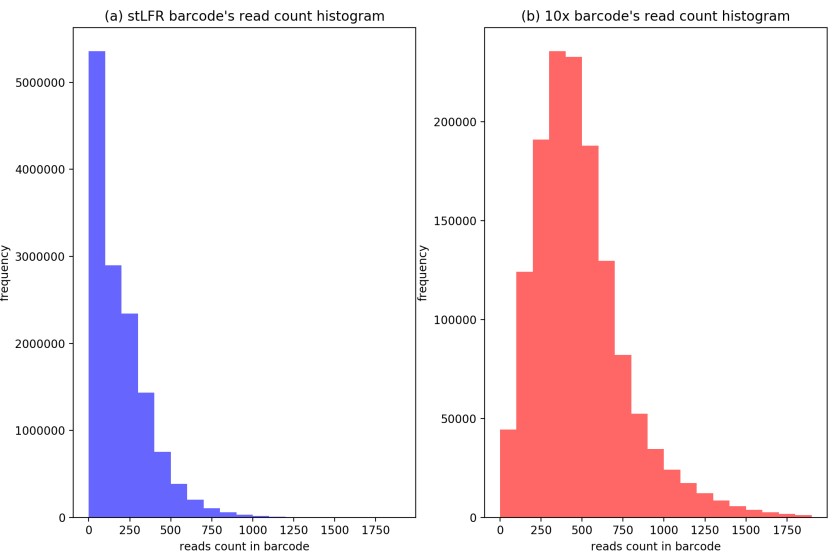

**Figure 4 The distribution of cluster's barcodes number of BGI's stLFR and 10× Genomics dataset.** (A) Generated by IterCluster with parameter -g 300, -k 2, -f 2 in BGI's stLFR dataset. (B) Generated by IterCluster with parameter -g 150, -k 5, -f 2 in BGI's stLFR dataset.

## IterCluster for de novo assembly

In the IterCluster result, there were enough reads data in an LFR cluster set to support de novo assembly. To improve the effect of the assembly, cluster false positive reads were removed using the FalseRemove module according to an average $k$-mer frequency for each LFR, as we assumed that the non-target LFR $k$-mer frequency was lower than a targets LFR. Then, we used SOAPdenovov2 (*Luo et al., 2012*) to assemble each IterCluster result set. For each subassembly result, we filtered contigs less than 1 kb because these were most likely sequences assembled by false positive reads. After that, we merged each assembly result together, and evaluated the results with QUAST (*Gurevich et al., 2013*).

## RESULTS

### IterCluster performance

To evaluate the cluster size after IterCluster in two dataset. We show the barcodes number distribution (Fig. 4) and reads number distribution (Fig. 5) of BGI stLFR datasets and 10× Chromium reads datasets.

To evaluate the performance of IterCluster on BGI stLFR datasets and 10× Chromium reads datasets, we plotted R-P to analyze the accuracy and recall of IterCluster. As can be seen from Figs. 6A, 6B IterCluster's accuracy and recall rate were high compared to baseline ($k = 1$). IterCluster performed better on BGI stLFR data than on 10× Chromium reads data. Figures 6C and 6E shows the performance of different parameters using the BGI stLFR data, where the $f$ parameter represents the lowest frequency threshold in the frequency selection model and the g parameter represents the unique $k$-mer number threshold between barcodes with overlap. Increasing the -$f$ and -$g$ values did not significantly

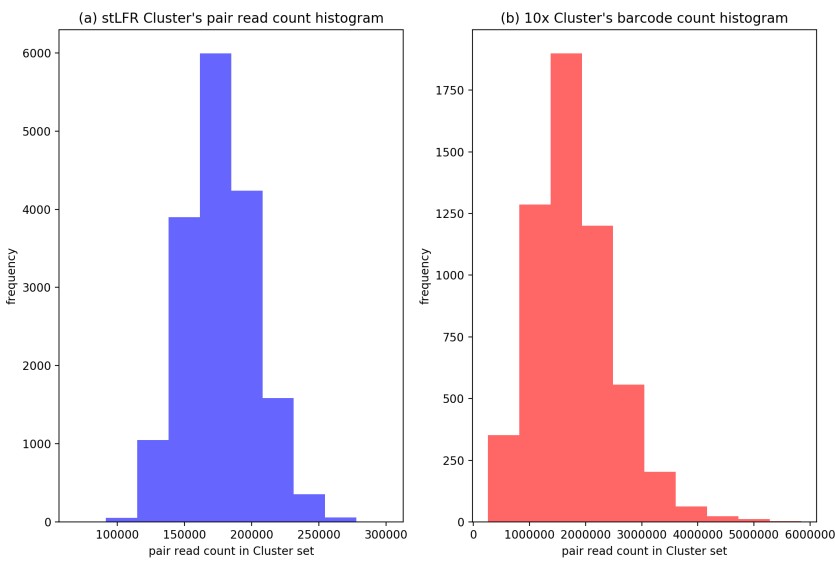

**Figure 5** **The distribution of cluster's reads number of BGI's stLFR and 10× Genomics dataset.** (A) Generated by IterCluster with parameter -g 300, -k 2, -f 2 in BGI's stLFR dataset. (B) Generated by Iter-Cluster with parameter -g 150, -k 5, -f 2 in BGI's stLFR dataset.

improve the accuracy of IterCluster, but it greatly reduced the recall rate. However, from Fig. 6F, we found that a slight increase in the -g value can improve the accuracy and recall rate using 10× Genomics Chromium reads data.

An important factor affecting the precision of IterCluster was the repeat sequences in genome. That is why we choose unique kmers as features. The $K$ value (the base number of each kmer) is an important parameter because it determines the specificity of a kmer. We thus studied IterCluster's performance with different $K$ values using BGI's stLFR data. Figure 7 shows that IterCluster can achieve higher precision and recall with larger $K$ values.

### $k$-mer frequency diversity selection model improve clusting

IterCluster needs to deal with two key issues in clusting: (1) how to determine the overlap between barcodes; (2) how to avoid false positives overlap caused by multiple fragments contained in the same barcode. These two issues are described in detail below.

In regard to the first issue, because overlap between different barcodes means that these barcodes will contain common unique $k$-mer, we use the number of common unique $k$-mer to measure the distance between barcodes. In the IterCluster, a user-specified common unique $k$-mer threshold (-g parameter) is used to determine whether two barcodes contain overlap. This threshold must be high enough that repeats and sequencing error do not result in false overlaps yet low enough that slight overlaps overlaps between fragments can be detected. This threshold can be estimated by the the distance distribution between all barcodes (Fig. 8A).

Another issue is caused by multiple fragment contained in barcode. Most of target barcodes carry a fragment from seed barcode area but carry fragments from other areas of the genome randomly. Therefore, after each cluster iteration, the feature of the seed

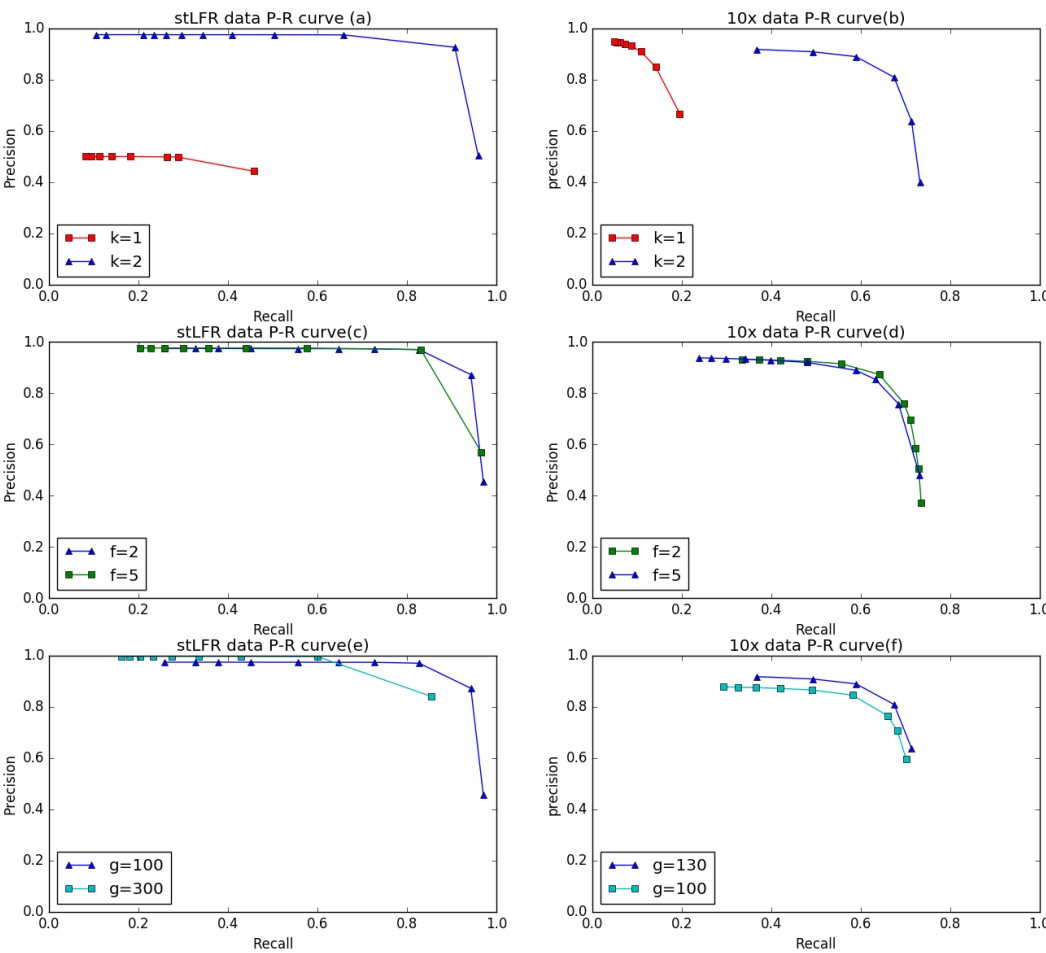

**Figure 6** **The preference of IterCluster on stLFR and 10× Genomics Chromium read with different parameter.** (A) Generated by IterCluster with different -k parameter in BGI's stLFR dataset. (B) Generated by IterCluster with different -k parameter in 10× Genomics Chromium read dataset. (C) Generated by IterCluster with different -f parameter in BGI's stLFR dataset. (D) Generated by IterCluster with different -f parameter in 10× Genomics Chromium read dataset. (E) Generated by IterCluster with different -g parameter in BGI's stLFR dataset. (F) Generated by IterCluster with different -g parameter in 10× Genomics Chromium read dataset.

barcode area (unique $k$-mer) need to be updated to prevent non-target area's unique $k$-mer from being used as feature for the next iteration of clusting. When choosing feature unique $k$-mer, IterCluster uses a unique $k$-mer selection model based on $k$-mer frequency diversity. The key idea is that after each iteration of the cluster, the unique $k$-mer frequency from seed barcode area is higher than the unique $k$-mer frequency from non-seed barcode area, because target barcode always has an overlap with seed barcode. Base on frequency diversity, IterCluster provides a user-specified minimum frequency threshold (-f parameter) to filter the unique $k$-mer from non-seed barcode area. This threshold can be estimated from the unique $k$-mer frequency distribution after each clusting iteration (Fig. 8B).
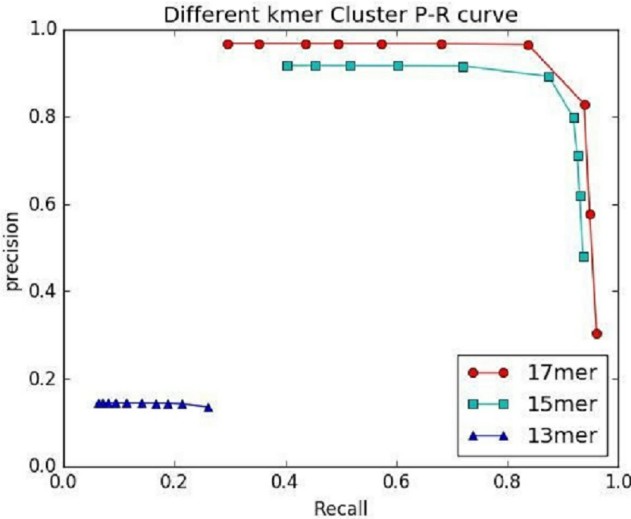

**Figure 7** **The preference of IterCluster using stLFR data with different *K* values.**

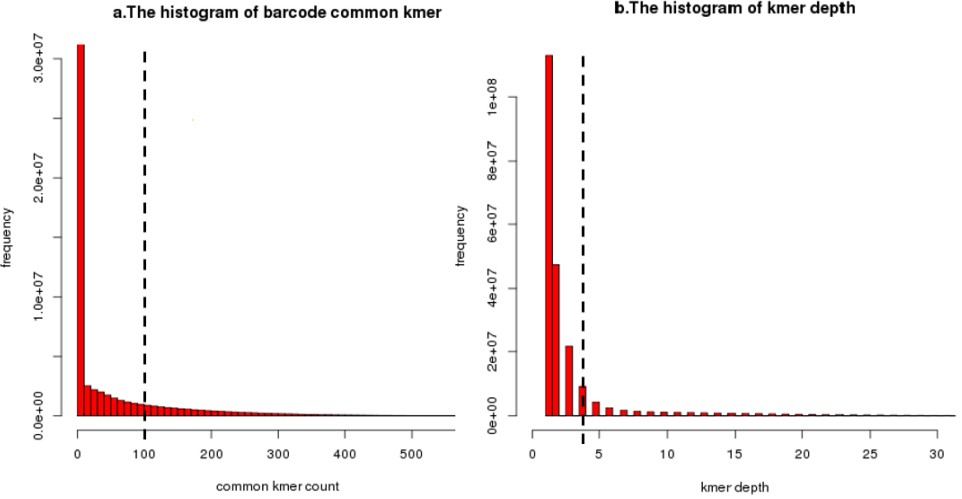

**Figure 8** **An illustration of common unique *k*-mer threshold estimation and minimum unique *k*-mer frequency estimation.** (A) A histogram of common unique *k*-mer number for 10,000 randomly selected seeds with the user-specific threshold shown as a dashed line for BGI's stLFR data. All connection which common unique *k*-mer up to 100 was considered to be a true overlap between barcode. The connection which common unique *k*-mer lower than 10 has been calculated but filtered. (B) A histogram of unique *k*-mer frequency for 100 randomly selected clusters' first iteration for BGI's stLFR data. The user-specific threshold was shown as a dashed line. All unique *k*-mer with frequency high than 3 within a cluster's first iteration was considered to be feature unique *k*-mer for next iteration. The unique *k*-mer which frequency is too high (>100) has been filter on this histogram.

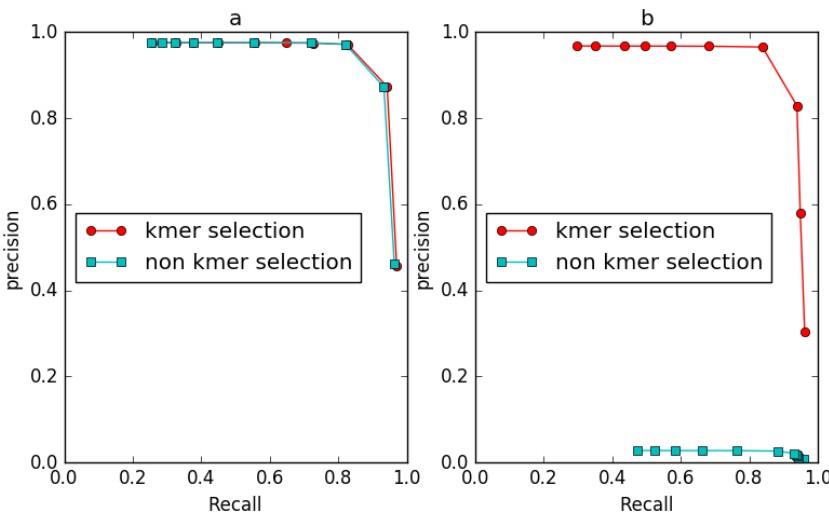

**Figure 9 The comparison of IterCluster's performance with and without *k*-mer frequency diversity selection model for BGI's stLFR data.** (A) The iteration k was set to 2. (B) The iteration *k* was set to 3.

To evaluate the effect of the frequency diversity selection model in controlling false positives, we compared the performance of IterCluster with or without this model (Fig. 9). When iteration $k = 2$, the number of unique $k$-mers in non-seed fragment area is still very small, and the frequency selection model has slight effect on IterCluster. However, when the number of iteration increased ($k = 3$), the false positive rate of IterCluster without frequency diversity selection model is significantly increased. This shows that the frequency selection model can control false positive rate with the number of iteration increased.

## Runtime and memory cost

IterCluster's runtime and memory cost largely depends on the parameter $K$, the kmer size used to calculate the adjacency matrix. The runtime and RAM usage for IterCluster with different $K$ values is shown in Table 4. The time and memory cost of IterCluster mainly stem from the adjacency matrix construction step of this algorithm. The larger the $K$ value, the more kmer numbers, and the more time is needed to calculate the relation between all kmers. However, changes in K do not greatly affect memory consumption. Although a larger $K$ value means a larger hash is needed to map all of the kmers, for each single kmer, the number of barcodes containing the kmer decreases. Therefore, the consumption of memory is stable and only decided by the amount of data. IterCluster is a highly parallel tool, both in the matrix build step and in its clustering step. We demonstrated that IterCluster can finish a 50-fold human whole genome dataset in 4 days with 50 threads.

## IterCluster improves de novo assembly using a divide-and-conquer strategy

We used IterCluster to improve the de novo assembly of human genome stLFR data from BGI via a divide-and-conquer strategy. First, we ran IterCluster on whole human genome stLFR data with default parameters. For each barcode cluster, we filtered the

**Table 4** **Runtime and memory cost of IterCluster with different _K_ values.** IterCluster run on BGIs stLFR data with default parameters and different _K_ values. The thread number was set to 50.

|  | Memory | Time |
|---|---|---|
| 17-mer | 327G | 73 h |
| 15-mer | 333G | 37 h |
| 13-mer | 326G | 12 h |

**Table 5** **Comparison of the performance of de novo assembly between divide-and-conquer strategy and normal method.**

|  | Contigs | | | Scaffolds | | |
|---|---|---|---|---|---|---|
|  | N50(bp) | Cov.(%) | Misass.(#/sum) | N50(bp) | Cov.(%) | Misass.(#/sum) |
| Cluster | **11,937** | **90.67** | 4,859/30M | **17,188** | **91.00** | 6,236/58M |
| Uncluster | 7 127 | 88.45 | 1,626/9M | 13 216 | 89.49 | 2,819/ 27M |

false positive reads with the FalseRemove module which was also added to our pipeline and assembly tasks were performed independently using SOAPdenovo2 (_Luo et al., 2012_). After the subassembly of each cluster, the assembly results from different clusters were integrated together. We call this assembly strategy "divide-and-conquer" or subassembly. In comparison to a baseline SOAPdenovo2 assembly using all reads (Table 5), assembly with this divide-and-conquer strategy achieved a longer contig length, as measured by the N50 length metric, of 11.9k, representing a 67% increase over the initial baseline achieved using SOAPdenovo2 assembly from all of the reads. Assembly with our divide-and-conquer strategy achieved a 30% increase in scaffold creation (N50 = 17.1K) and a slight improvement in genome coverage. This suggests that IterCluster uses more of the long-range information present in stLFR data to break down a complex chromosomal genome into multiple simple sub-sections.

## DISCUSSION

Supernova is an efficient and excellent assembly algorithm specifically designed for use with 10× Genomics Chromium reads data that capitalizes on a de Bruijn graph (DBG) strategy, leading to assembly of contigs with N50 up to 100 kb (_Weisenfeld et al., 2017_). However, stLFR data cannot be assembled using supernova, because supernova only accepts the unique characteristics of 10× Genomics Chromium reads, such as the barcode number and the amount of data for each barcode. The improvement of the contig N50 using a divide-and-conquer assembly strategy suggests that IterCluster can reduce the complexity of de novo assembly through local assembly, and thus provides a platform for de novo assembly using stLFR data. In contrast to TSLR technology, IterCluster capture reads from fragment overlap using a seed and achieves local assembly, so it does not actually improve the local coverage of each fragment and the captured reads contain all of the same haplotype, making it unsuitable for haplotype phasing.

## CONCLUSIONS

We have introduced IterCluster, a novel algorithm providing an optional solution to the barcode cluster problem. IterCluster provides a conservative solution for exploring the potential relationships between barcodes and realizes barcode clustering based on a $k$-mer frequency selection model. Relying on IterCluster, the barcodes of a certain area in the genome can be enriched, thus achieving de novo assembly by a divide-and-conquer strategy.

The current version of IterCluster provides reasonable performance, but this jumps up to 350 cpu-hours for clustering barcode from some large-scale datasets. Although a time-efficient way to generate adjacency matrices have been developed, the size of an intersection between optional barcodes and a seed must be recalculated, because the feature $k$-mers are updated after each clustering round. A large performance bottleneck is that the recalculation method is naive and precise. Future versions of IterCluster will employ local sensitive hashes (*Charikar, 2002*) to improve the performance.

In contrast, IterCluster uses the same relation threshold (*-g*) to capture target barcodes, but using the same relation threshold on each cluster round is unbefitting and future versions will consider more appropriate criteria for capturing barcodes. Also, the same relation threshold was used to measure the relation of each barcode. However, the number of reads and fragments on each barcode are different, and a more appropriate mathematical model is needed to measure the relation between barcodes.

Overall, we believe that IterCluster is a significant tool for taking advantage of LFR data. LFR data has the potential to improve de novo assembly and to allow local assembly.

## ACKNOWLEDGEMENTS

We would like to acknowledge the ongoing contributions and support of MGI's algorithm team, in particular Jingbo Tang and Weihua Huang for their suggestions towards LFR data analysis and algorithm performance optimization. We would also like to thank BGI-Shenzhen for providing the stLFR data and information about its characteristics. We thank LetPub for its linguistic assistance during the preparation of this manuscript.

### Funding

This work was supported by the Shenzhen Peacock Plan (NO.KQTD20150330171505310). There was no additional external funding received for this study. The funders had no role in study design, data collection and analysis, decision to publish, or preparation of the manuscript.

### Grant Disclosures

The following grant information was disclosed by the authors:
Shenzhen Peacock Plan: KQTD20150330171505310.

## Competing Interests

Jiancong Weng, Tian Chen and Yinlong Xie are employees of MGI. Xun Xu is a group leader of BGI-shenzhen. Brock A. Peters and Radoje Drmanac are scientists at BGI-shenzhen.

## Author Contributions

- Jiancong Weng performed the experiments, prepared figures and/or tables, authored or reviewed drafts of the paper, develop the software, and approved the final draft.
- Tian Chen analyzed the data, prepared figures and/or tables, and approved the final draft.
- Yinlong Xie and Gengyun Zhang conceived and designed the experiments, authored or reviewed drafts of the paper, and approved the final draft.
- Xun Xu, Brock A. Peters and Radoje Drmanac conceived and designed the experiments, prepared figures and/or tables, coordinated the study, and approved the final draft.

## Data Availability

Data is available at GitHub: https://github.com/JianCong-WENG/IterCluster.

## Supplemental Information

Supplemental information for this article can be found online at http://dx.doi.org/10.7717/peerj.8431#supplemental-information.

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
