# Peer review of "IterCluster: a barcode clustering algorithm for long fragment read analysis"

_PeerJ, doi:10.7717/peerj.8431_

## Round 0.1 · original submission · Major Revisions

The reviewers have raised several important, major issues with the manuscript that need to be addressed before the paper can be considered again for publication. In addition to directly addressing all reviewers' concerns, recent literature must be discussed in Introduction and Results and appropriate methodological comparisons should be made with similar available algorithms. Also, the manuscript needs extensive editing and proofreading for language and grammatical mistakes.

Reviewer 1 ·

Basic reporting

1) The manuscript needs serious editing and proofreading to be more readable.
The main idea of this paper was already presented in the following manuscript: Minerva: An Alignment and Reference Free Approach to Deconvolve Linked-Reads for Metagenomics (https://www.biorxiv.org/content/early/2018/07/21/217869)

The similarities and differences of this method with Minerva should be addressed and discussed:

2) In addition to comparison with Minerva, there should be a comparison with alignment-based method since a human reference genome or an assembly is available.

Experimental design

1) It’s not clear how the -g value is determined?

2) What is exactly a divide-and-conquer assembly?

3) There should be a test on full human genomes.

Validity of the findings

1) stLFR data characteristics are not discussed. All the way through the results section, it seems to me that it’s similar to the 10x Genomics data, but then the authors could not run supernova on this data set? Why? what are the unique features of sfLFR?

Additional comments

The stLFR data is a new linked-read technology but there is not really a discussion about it. This should be discussed in details.

Reviewer 2 ·

Basic reporting

- In this paper, authors presented a method for clustering the barcoded long fragment reads generated from linked-read sequencing technologies. The proposed method is based on alignment-free paradigm and utilizes the k-mer content of reads to cluster them into target regions in the genomes. To evaluate the performance of their method, authors used two datasets from BGI stLFR and 10X genomics chromium read data. Partitioning reads with a single barcode into clusters that correspond to a single long fragment of DNA is one particular issue common to all applications of the linked-read technology. The problem and idea of clustering long fragment range data is very interesting and will be widely encountered in the area. However, the manuscript has a number of shortcomings, ranging from Introduction and Method details to the presentation of results and conclusions. There are several major concerns to need a substantial amount of time to be resolved as follows. Here are several examples falling into each category.

- There are so many typos/mistakes in English of the manuscript. So, I suggest asking a professional proofreader to improve the English of the paper.

- Authors should modify Introduction/literature to include recent studies on clustering linked read/ read cloud data. For example, there is a new tool Minerva recently published for barcode deconvolution of linked-read/read-cloud data: "Minerva: An Alignment and Reference Free Approach to Deconvolve Linked-Reads for Metagenomics"

Experimental design

- Have authors considered and tested other clustering approaches instead of MCL?

- Runtime and memory efficiency are two important features on clustering large-scale linked-read/read-cloud datasets. I don’t see any runtime and memory usage performance analysis of the proposed method. Author should provide comprehensive analysis on these features.

- There is no justification on choosing the k-mer length from the statistics of k-mer coverage profile. How do the authors choose the optimal value of k for the comparison? I would also like to see the runtime/memory performance for different values of k.

- Based on the hash map data structure presented in Fig 2, the required memory for large-scale long-range data could be very large and may not fit on a single compute node. Therefore, the scalability of the proposed algorithm to large-scale datasets could be an important challenge.

- There is no specification for the datasets used in the experiments. Author should provide a table for dataset specification. Further, the URL links to the datasets used in the experiments are not working.

- Author should provide a help manual for running IterCluster in Supplementary Data and on the GitHub page. They also should explain all details and options for running the tool.

Validity of the findings

- Results section is too preliminary unfortunately to convince that the proposed method is a significant tool to take advantage of long fragment range data to improve downstream results such as de novo assembly. Using the proposed method on just chromosome 19 data for evaluating the assembly performance improvement is not robust enough for validation of the proposed tool. So I recommend to use a larger dataset for this section.

- Runtime and memory usage should also be reported in the Results section as a table.

- It would be interesting to see the performance comparison of IterCluster and Minerva in clustering the barcoded long fragment reads.

---

## Round 0.2 · Major Revisions

Although the reviewers have raised interest in the method and believe that the manuscript has improved in its revision, they still have substantial major concerns about the manuscript that must be addressed. Importantly, the reviewers could not install and execute the pipeline. Moreover, there are issues with performance comparison, lack of run-time and memory usage analysis, and the presentation of the underlying mathematical concepts of the algorithm.

Reviewer 1 ·

Basic reporting

Overall Evaluation
The authors seem to have produced a fine tool with a meaningful use case and an interesting approach. With the revision, this paper is likely to be publishable. However, the manuscript as is still fails to clearly describe how the algorithm works and how it was evaluated.

In particular, the authors do not include an overall description of their algorithm in the methods section. This is by far the biggest issue with the current manuscript and is the root of most other issues.

In their response, the authors protest that making modifications requested by the reviewers would take excessive compute time. While this may be justified for Minerva which is designed for microbial samples (see notes below) it is not justified for alternate clustering algorithms or for comparisons to the human reference genome. The current paper does not broadly enough establish a comparison between the presented technique and other reasonable methods.

Major Points
The authors do not clearly explain their algorithm. The section describing the algorithm includes many section explaining subcomponents of the algorithm but it is not obvious how these components are used together. The authors should add a description of the algorithm overall.

As requested before, the authors should compare their method to direct alignment with the human reference genome. We understand that this method does not require a reference genome but the comparison is still necessary to establish a performance baseline.

The authors confude programmatic parameters (i.e. -g) with the mathematical concepts they represent. The main text should not include references to programmatic parameters and should clearly explain what user-defined values can be supplied to the algorithm and how these are chosen.

The description of Markov Clustering (lines 195-206) explains the issues associated with clustering in this context but it does not explain why Markov clustering is useful in this context. This description is critical as Markov clustering is a family of related algorithms. The authors must both define why Markov clustering is useful in this context and explain how Markov clustering works in their method.

The authors seem to present their issues with 10x reads as failures of the data rather than their algorithm. The authors focus too much on the differences between 10x and BGI stLFR in their algorithms performance. This paper is absolutely not an evaluation of the differences between these two data types and should not be presented as such.

The Description of Data Analysis and Data Sources should be more clearly separated from the description of the algorithm. Though both of these are, arguably, methods sections the Data Analysis and Data Sources methods would be more naturally included with the results section since that is where they are relevant.

The Data Analysis section is far too abbreviated. The authors should much more clearly explain their methodology and evaluations. In particular, the authors should:
• Separate data filtering and quality control and justify why these parameters are different for the two datasets.
• Move all commands and parameter lists to the supplements. These choices should be explained in the text but merely presenting the parameters does not justify the choice of filter. Few readers will be intimately familiar with the exact interface of the tools being used.
• Because the overall algorithm is not clearly described it is nearly impossible to follow the exact steps used in the evaluation. After adding an overall description of the algorithm the authors should directly reference which steps match which parts of the algorithm
• More clearly define what indicates a true-positive and false-positive in this context.
• Show the cluster size.
• Include comparisons to other reasonable techniques (like using the reference genome or other clustering algorithms.)

Minor Points
The authors have misspelled ‘clustering’ as ‘clusting’.

Not always clear what the authors intend you to look for in figures. It would benefit readers if figure references included ‘hints’. I.e. the reference to figure 1C in line 109 is a reference to a cartoon showing how false positives can occur but from context I expected a figure showing the false positive rate

The caption on figure 3 is not informative. It’s not clear what k, f, and g are from the figure.

The choice to filter reads which span fewer than 500bp (line 232) feels aggressive. How many reads does this choice remove?

The use of precision/recall to measure cluster accuracy could be supplemented by other simpler metrics.

Comments on Minerva
- Minerva is designed for metagenomic samples which are typically much smaller than human genomes. The authors should mention this.
- The manuscript would benefit from a more clear distinction between barcode deconvolution and barcode clustering as this seems to be confusing
- When running on a single chromosome (19) it is likely that most of the barcodes are already deconvolved which is likely why Minerva produces little output

Experimental design

no comment (see 1. Basic reporting)

Validity of the findings

no comment (see 1. Basic reporting)

Additional comments

no comment (see 1. Basic reporting)

Reviewer 2 ·

Basic reporting

The authors have addressed some of my comments in the previous report but the manuscript still needs more improvement.

Experimental design

I tried several times to build the binaries and perform some tests and comparison of the proposed method but due to some hardcoded environmental variables in the scripts I was not able to do so. Authors should resolve all errors and bugs in building the binaries.

$ sh make.sh
make.sh: line 3: /ldfssz1/MGI_ALGORITHM/assembly/huangweihua/bin/cmake-3.9.1/bin/cmake: No such file or directory
Errors executing CMake. Fix errors and run again.

$ sh build.sh
CMake Error: The current CMakeCache.txt directory /var/tmp/IterCluster/IterCluster/CMakeCache.txt is different than the directory /ldfssz1/MGI_ALGORITHM/assembly/wengjiancong/My_software/IterCluster-MCL where CMakeCache.txt was created. This may result in binaries being created in the wrong place. If you are not sure, reedit the CMakeCache.txt
make: /ldfssz1/MGI_ALGORITHM/assembly/huangweihua/bin/cmake-3.9.1/bin/cmake: Command not found
make: *** [cmake_check_build_system] Error 127



Modify all make scripts such as makefile.cmake, build.make, make.sh, Makefile, CMakeCache.txt, … and change the hardcoded vars like:

CXX=/share/app/gcc-5.2.0/bin/g++
DBOOST_ROOT=/ldfssz1/MGI_ALGORITHM/assembly/huangweihua/bin/boost_1_46_1

Validity of the findings

The experimental results are not complete.
- What would be the runtime and memory usage of the proposed method for larger k values? I would also like to see the runtime and memory usage of IterCluster for k values like k=20, 30, and 40.
- How scalable is the proposed method on 50X human whole genome dataset with different number of threads?

Additional comments

There are some typos in the text:
Line 179: frequency --> frequencies
Lines 262, 263, 277, 283: clusting --> clustering
Line 273: fragment --> fragments

Further: change all “kmer” to “k-mer” to be consistent.

There is no link to the GitHub page of IterCluster in the main text. Please add https://github.com/WengJianCong/IterCluster to the manuscript.

---

## Round 0.3 · Minor Revisions

I am happy to let you know that the reviewers have found the revised manuscript and your response to their comments mostly satisfactory. However, a few minor issues have been raised, which should be addressed before acceptance of the paper.

Reviewer 1 ·

Basic reporting

See the attached PDF

Experimental design

See the attached PDF

Validity of the findings

See the attached PDF

Additional comments

See the attached PDF

Annotated reviews are not available for download in order to protect the identity of reviewers who chose to remain anonymous.

Reviewer 2 ·

Basic reporting

My comments for the previous version have been addressed.

Experimental design

I have no further comments.

Validity of the findings

I have no further comments.

---

## Round 0.4 · accepted · Accept

Congratulations on the acceptance of your paper to PeerJ.